# Does women empowerment associate with reduced risks of intimate partner violence in India? evidence from National Family Health Survey-5

Rakhi Ghoshal[1], Priti Patil[2], Anita Gadgil[3], Priyansh Nathani[4], Prashant Bhandarkar[2], Dnyaneshwar B. Kale[5], Nobhojit Roy [6,7] *

1 Gender Equality Centre, CARE India, New Delhi, India, 2 Department of Statistics, BARC Hospital, Mumbai, India, 3 Department of Surgery BARC Hospital, Mumbai, India, 4 Hinduhridaysamrat Balasaheb Thackeray Medical College and Dr Rustom Narsi Cooper Municipal General Hospital, Mumbai, India, 5 Indian Institute of Health Management Research, Rajasthan, India, 6 Department of Global Public Health, Karolinska Institutet, Sweden, 7 The George Institute for Global Health, New Delhi, India

* nobhojit.roy@ki.se

## Abstract

### Background

Women empowerment is commonly believed to be an important factor affecting a woman's likelihood of facing violence from her intimate partner. Even as countries invest in policies that aim to strengthen women empowerment, studies show that increase in women empowerment does not necessarily decrease intimate partner violence (IPV) against them. Against this paradox, the present study seeks to understand the specific empowerment components that associate with IPV against women in India. It also studies the state-level distribution of the different types of IPV.

### Methods

The study analyses state-level data from the National Family Health Survey, India (2019–21). A total of 72,056 women responded to the domestic violence questionnaire. The Dimension Index (DI) was used to compute composite scores for Women Empowerment and for IPV to rank states and Union Territories. The correlation between Women Empowerment and IPV scores was determined using Spearman's rank correlation coefficient.

### Results

The state of Karnataka had the highest composite score of IPV and also showed the highest burdens of physical, sexual and emotional IPV, while Lakshadweep had the lowest burden. Physical IPV was the most common form of IPV for most states across the country. The states in the western part of India had reduced burdens for all three types of IPV. Three specific components of empowerment, viz. household decision-making and mobile phone ownership significantly associated with reduction of all three types of IPV. Hygienic menstrual practices strongly associated with reduction of sexual and emotional IPV. However, property

**Data Availability Statement:** All data accessed by this study is publicly available for download at http://rchiips.org/nfhs/factsheet_NFHS-5.shtml.

**Funding:** The author(s) received no specific funding for this work.

**Competing interests:** The authors have declared that no competing interests exist.

ownership of women increased risks of all three types of IPV, while employment had no significant association with any type of IPV.

## Conclusions

The study found no significant reduction in overall IPV with improvement in women empowerment. However, it identifies components of empowerment that associate with IPV. Household decision-making, ownership of mobile phones, and hygienic menstrual practices associated with a lowered risk. By contrast, owning property increased the risk. The findings of this study would inform future research and intervention that aim to strengthen specific components of women empowerment in India and other low-and-middle-income countries.

## Background

Violence inflicted by one's intimate partner or the spouse (Intimate Partner Violence) is a global problem with severe physiological and psychological health consequences [1]. Intimate Partner Violence (henceforth, IPV) can be physical, sexual, emotional and psychological in expression [2]. Globally, one in three women face IPV at least once in their lifetime [3]. While there are several factors associated with IPV [4] one of the most examined factors is women empowerment [5]. The standard components of women empowerment include level of education, ability to take household decisions and access to material resources [5,6]. Empowerment is commonly assumed to reduce the incidence of IPV in a woman's life [7]. The United Nations, in the Beijing Declaration, endorsed women's economic empowerment as a protective factor for IPV [8].

Certain countries including India, have put in place policies and legislations that prioritise strengthening the different components of women empowerment. Legal provisions and Acts such as the Maternity Benefit Act (1961), the Equal Remuneration Act (1976), Protection of Women from Domestic Violence Act (2005), and the Prohibition of Sexual Harassment of Women at Workplace Act (2013) among others, are aimed to empower women across different spheres of life. When observed longitudinally, women empowerment has increased in India [9]. However, evidence shows that increase in empowerment has not necessarily translated to a reduction in incidence of IPV. Evidence shows that components of empowerment such as employment, actually increase the risk of IPV [4,6,8]) up to nine times [10]. Data over a decade found that women with higher levels of education, and with increased economic control over household resources were more vulnerable to IPV compared to women who had lower levels of education and reduced economic control at the household level [11]. Scholars have argued that while women who are not financially empowered face IPV because of their vulnerability and dependence on their partners, women who are financially autonomous "may promote insecurity and feelings of economic inadequacy" triggering the traditionally dominant partner, i.e. the man to perpetrate violence [12].

In the India context, studies on women empowerment and IPV have also found no significant reduction of IPV against women when empowerment increased [8,13]. A study [14] comparing the prevalence of IPV between two rounds of a household level cross-sectional health survey (2005–06 and 2015–16) found that women empowerment was not protective against IPV; empowerment in fact increased the risk. A longitudinal study [15] from Bangalore, India found that once women took up employment their likelihood of IPV increased by up to 80% compared to women who continued to be unemployed. In-depth data on the association

between women empowerment and IPV in India is building up; scholars have identified gaps such as the need to study association of specific components of women empowerment with different types of IPV [8] as well as the need to study regional variations for this association [16].

Against this background, the present study aims to address both these identified gaps; it seeks to evaluate the association of specific components of women empowerment with different types of IPV against women in India. The study also analyses a state-level distribution of the different types of IPV.

# Method

## Design and setting

This study offers a state-level analysis of the association between women empowerment and risk of IPV. It does a secondary analysis of cross-sectional data that was collected as part of the household level survey, the National Family Health Survey-5, between 2019 and 2021. The National Family Health Survey (NFHS) is modelled on the Demographic and Health Survey (DHS), and captures a wide range of health parameters, including data on Women Empowerment and Domestic Violence. The data is captured at the state and district levels, and supports policymakers, implementors and administrators to design implementations that are informed by evidence. The 5th round of the NFHS is the latest round and includes representative urban and rural household samples from all 28 states and 8 Union Territories (U.T.) across the country.

## Sample size

In this round of the NFHS data was collected from 636,699 households which included 724,115 women and 101,839 men. The Domestic Violence questionnaire module was administered to married women in the age group 18–49 years. Till the fourth round of the survey (2015–16), domestic and spousal violence/IPV questions were asked to married women in the age group 15–49 years. From the current fifth round, the lower limit of the age range was increased to 18 years. This was done to avoid the legal mandate that binds any person with the knowledge of child sexual abuse to report it to the authorities (as per the Protection of Children from Sexual Offences, 2012). Since under the changed law, a woman under the age of 18 years is legally a minor, any information about her sexual life, even if with her 'husband' would require reporting to the authorities. However, since underage marriage is widespread in India, not including all married women aged less than 18 years also implies that there is a loss of we also lost out on a significant volume portion of data.

To adhere to ethical guidelines, only one eligible woman per household was interviewed. Special weights were applied to account for the selection of only one woman per household and to ensure that the subsample related to domestic violence was representative at the national level. A multistage sampling approach was used to select the households for the survey. A uniform sample design was adopted to represent the sample at the national, state/union territory, and district levels. It stratified districts into urban and rural areas. Each rural stratum was divided into smaller substrata based on village population and the percentage of scheduled castes and tribes (SC/ST). Village clusters were selected as Primary Sampling Units (PSUs). In urban areas, Census Enumeration Blocks (CEBs) were selected as PSUs, sorted by the percentage of SC/ST population. Within these PSUs, 22 households were systematically chosen from a mapped and listed household database. A total of 30,456 PSUs were selected from 707 districts in NFHS-5. A total of 72,056 women provided responses to the domestic violence module. Weighting is applied to account for sampling complexities and achieve accurate population-level estimates.

### Independent variables: Women empowerment

We considered Women Empowerment as an independent variable. Empowerment was further disaggregated by the NFHS as, percentage of women (a) participating in major household decisions, (b) who worked during last year and were paid in cash, (c) who own house and/or land, (d) have an own bank account, (e) who have an own mobile phone, and (f) who use hygienic methods of protection during menses.

### Outcome variable: Women facing any IPV

IPV faced by ever-married women was further disaggregated into physical, sexual and emotional IPV. Percentage of women who faced either of the three types of IPV was the dependent variable and was calculated per state/U.T. The questionnaire for the Domestic Violence schedule is available as a S1 Table.

### Statistical analysis

Statistical analysis was carried out using Microsoft office Excel-2019 (for calculating Dimension Index), R Studio (for data visualization), and SPSS version 22.0 (for statistical techniques) (SPSS Inc., Chicago, IL, USA) for windows for doing statistical analysis. NFHS Data factsheet were extracted in comma separated value (CSV) file format which were further summarized for the analysis. Heat-map-based visualization is done using opensource technology R, which is a programming language and environment for Statistical computing and graphics. We used ggplot2 package of R for this purpose in the R studio for Windows. We used the Dimension Index (DI) measures developed by Iyengar and Sudarshan [17] Click or tap here to enter text. The spatial aspects of the development are used to measure the level or stage of the state's development. This method was preferred since it requires less assumptions compared to other methods such as Principal Component Analysis. The Dimension Index (DI) was calculated separately for each state and UT for all the independent variables.

$$Dimension\ Index\ (DI) = \frac{Actual\ Value\ of\ the\ Indicator - Minimum\ Value}{Maximum\ Value - Minimum\ Value}$$

For e. g. DI for the state of Bihar for physical IPV is calculated as:

$$DLBihar = \frac{Physical\ Violence\ \%\ for\ Bihar - Minimum\ Physical\ Violence\ \%\ across\ all\ states\ and\ UTs}{Maximum - Minimum\ values\ of\ the\ Physical\ Violence\ across\ all\ states\ and\ UTs}$$

The DI value lies between 0 and 1. A greater value indicates a higher rate. For any of the component of Women Empowerment, the closer a value is to 1, the better women empowerment is for that state. When calculating for IPV, the closer a value is to 1, the bigger the burden of IPV is for that state. The DI calculated composite scores for Women Empowerment and also for IPV. The composite score is the addition of the DI of respective indicators. The states and UTs were ranked according to the score.

Spearman's rank correlation coefficient is used to find the association between Women Empowerment and IPV composite scores. The standard convention of Dancey and Reidy [18] was used for the interpretation of Spearman coefficient ($\rho$) values and grouped the association as either weak ($\rho < 0.30$), moderate ($\rho$ between 0.30 to 0.39), or strong ($\rho \geq 0.40$). This correlation was estimated using SPSS [21] Any p-value less than 5% was considered statistically significant.

### Ethical considerations and data quality

The NFHS data is available for public download through the data distribution system of the Demographic and Health Survey (DHS). The study protocol for the NFHS is adapted from the DHS which is reviewed and has been approved by the United States Centers for Disease Control and Prevention. Further, for the NFHS, the ICF International Institutional Review Board and the International Institute for Population Sciences (IIPS) Institutional Review Board reviewed and approved the protocol. All personnel conducting the household survey and administering the tool were trained in standard research protocols including those pertaining to safeguarding vulnerable research subjects. In alignment with the World Health Organization's (WHO) ethical guidelines [19] on conducting surveys with survivors of domestic violence, only one eligible woman per household was selected for interview on Domestic Violence. Participant information sheet was distributed to each respondent in the appropriate language, and informed consent of all participants was taken during the survey. Dummy questionnaires were used to engage other female/ adult members so that the privacy of the respondent was not compromised. If privacy concerns arose and interview could not be attempted or concluded the interview was not conducted. Additional details on the protocol are available at https://dhsprogram.com/Methodology/Survey-Types/DHS.cfm.

## Results

Karnataka, with its highest composite score of IPV, shows up the highest burden for each type of IPV. We found that physical IPV is the most common form of IPV for most states (Fig 1). The Western states of India showed up with lighter burdens for all three types of IPV.

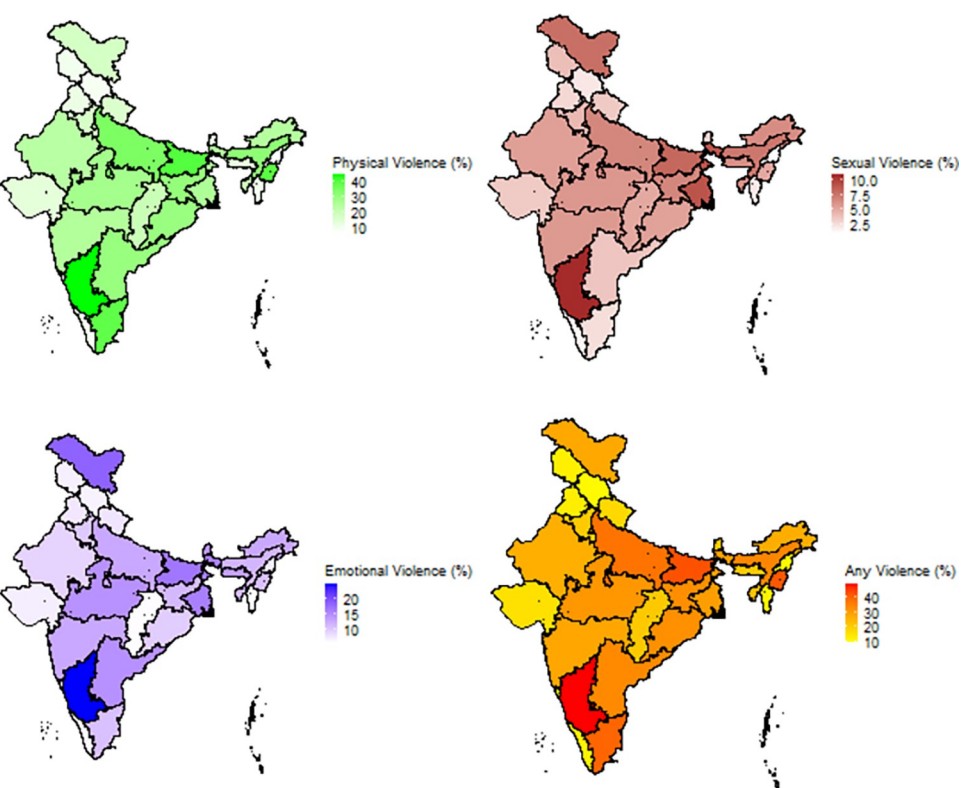

**Fig 1. State-wise heatmaps of each type of IPV and combined IPV.**

**Table 1. Abridged state level composite scores for women empowerment and IPV.**

| State | Intimate Partner Violence (from highest to lowest) | Women Empowerment |
|---|---|---|
| Karnataka | 3.0 | 3.73 |
| Bihar | 2.31 | 1.77 |
| Telangana | 1.95 | 4.21 |
| Madhya Pradesh | 1.72 | 1.61 |
| Puducherry | 1.03 | 4.95 |
| Goa | 0.76 | 4.21 |
| Lakshadweep | 0 | 2.85 |

In Table 1 we present the composite IPV scores (ranked from highest to lowest) versus the corresponding composite women empowerment score. In the range of composite scores for IPV, the lowest possible was zero, and belonged to Lakshadweep while the highest of three belonged to Karnataka. In other words, Karnataka has the highest burden of IPV while Lakshadweep has the lowest burden. However, when we considered the composite scores of women empowerment alongside each state, we found that empowerment was highest in Puducherry (4.95), closely followed by Telangana and Goa, both with a score of 4.21. Women empowerment scores were lowest for Madhya Pradesh (1.61), closely followed by Bihar (1.77). Women empowerment scores for Karnataka (which has highest IPV burden) was 3.73, and Lakshadweep (which has the lowest IPV burden) was 2.85. (S2 Table presents complete table of composite scores for women empowerment and IPV).

Table 2 studies the association of all three types of IPV, and also combined IPV, with specific components of women empowerment. As discussed in the Methods above, associations can be positive or negative, and classified as 'weak' for ρ <0.30, as 'moderate' for ρ between 0.30 and 0.39, and strong for ρ ≥ 0.40. The table shows that when considered overall, women empowerment has no significant association with IPV. However, empowerment has a strong negative association with only sexual IPV. Household decision-making is strongly associated with a reduction of overall IPV and also each type of IPV. Ownership of property had a strong association with increased risk of IPV, and particularly of emotional IPV. Ownership of mobile phones strongly associated with a reduction of IPV, and particularly of physical and sexual IPV. Hygienic menstrual practices strongly associated with reduced IPV, particularly of sexual and emotional IPV. Employment did not associate strongly with any type of IPV.

## Discussion

This study analysed the association between specific components of women empowerment and types of IPV in India. The empowerment components of household decision-making and

**Table 2. Association of IPV with various independent parameters using Spearman's rank correlation.**

| Spearman Rank Correlation Coefficient | Overall IPV | Physical violence | Sexual violence | Emotional Violence |
|---|---|---|---|---|
| Women empowerment | -0.243 (0.153) | -0.086 (0.62) | -0.433 (0.008) | -0.188 (0.272) |
| Decision making | -0.628 (0.000) | -0.402 (0.015) | -0.630 (0.000) | -0.706 (0.000) |
| Employed in last 12 months and earned in cash | 0.101 (0.557) | 0.206 (0.228) | -0.108 (0.530) | 0.179 (0.297) |
| Have own house/land | 0.442 (0.007) | 0.376 (0.024) | 0.328 (0.050) | 0.470 (0.004) |
| Having bank account | -0.065 (0.708) | 0.121 (0.481) | -0.16 (0.350) | -0.154 (0.369) |
| Mobile phone | -0.512 (0.001) | -0.530 (0.001) | -0.507 (0.002) | -0.342 (0.041) |
| Menstrual hygiene | -0.453 (0.005) | -0.261 (0.125) | -0.578 (0.000) | -0.415 (0.012) |

Figures indicate Spearman's rank correlation coefficient-ρ (p-value).

mobile phone ownership associated with a significant reduction of all three types of IPV; while hygienic menstrual practices associated with reduction of sexual and emotional IPV. [7].

The study found that overall women empowerment did not associate with any significant reduction of IPV. Employment had no impact on any type of IPV as well. Other studies have found that empowerment components such as employment or high levels of education of the woman do not associate with reduced IPV because "mainstream formal education does not emphasize self-protection against gender-based violence" [20]. In most cultures IPV is part of social norms, and as long as the pedagogical content does not train us to question this normalization or teach us methods of identifying and resisting violence, mere increase in access to education shall not by default reduce the burden of IPV. A study [21] from Nigeria found that when women in communities where female employment was not a norm, got employed and began to have their own earning, they were much more at risk of IPV than their unemployed counterparts. However, in communities where female employment was a norm, their risk of facing IPV did not reduce significantly. Despite this seeming impasse, Weber says [21] that women who are perceived as violating gender norms by taking up employment might be at increased risk in the immediate term, but, "in the longterm, [they] may serve as catalysts for changes of norms that ultimately improve employment opportunities for future generations of women".

The present study found that women empowerment had a significant association only with reduction of sexual IPV. This is in agreement with findings from other studies [22]: empowered women are more likely to have increased mobility or be employed, which would translate to their having reduced contact time with their husbands who are the perpetrators of sexual IPV [22] Also, empowered women are more likely to have increased bodily autonomy, implying that they would either have a reduced threat of sexual IPV or be better equipped to resist sexual IPV [23]. Since marital rape is yet not criminalized in India, it is important that women themselves become capable of resisting sexual IPV, and this is thus an encouraging finding of the study.

The present study also found that increase in household decision-making associated with reduced IPV in all three forms. IPV is a manifestation of abuse of structural power [24]; women in power inequitable intimate relationships are very likely to have little to no bargaining or decision-making power [25] within intimate relationships. Conversely, when women have increased decision-making power, they are likely in a more power equitable relationship. Such relationships are thus more likely to experience lowered risks of IPV. The household is the smallest unit of society and household decision-making is "an indicator of power and control within relationships" [13]. The present study found that states with higher proportions of women who had solo or/and shared household decision-making power also had lowered IPV burden. This is in sync with evidence from other low-and-middle-income countries. A study from Uganda [13] found that women are best protected against IPV when they made household decisions, and particularly when decision-making was shared with their husbands. A study from Ethiopia [26] and another from Philippines [12] presented similar arguments, viz. decision-making, when shared with the husband, reduced IPV even more than when done only by the woman. The present study, however, did not unpack decision-making as by the woman and shared with the husband.

As per the findings of the present study, ownership of mobile phones associated with a reduction of all three types of IPV. Besides a symbol of empowerment, mobile phones are also instruments of social connection. Women with mobile phones are more likely be connected with safe contacts and a support network. Mobile phones have the symbolic power to contribute to "mitigating women's fears, sense of isolation, loneliness, and boredom by helping them cope with confinement at home" [27]. Wide availability of mobiles phones translates to

improved financial independence, food security and dietary quality, better household decision-making power by women [27]. Pesando further discusses a study [27] that has shown how the use of mobile phones significantly decreased "both men's and women's tolerance for IPV and increased women's autonomy in mobility and economic independence". Pesando's study [27] of mobile phones and IPV across 10 LMICs found that women with mobile phones consistently faced reduced IPV risks. The study found that in households where women had their own mobile phones were likely to be households that had a better recognition of women's need and rights to share household material resources and have their own networks of friends and families [27].

The study found that women who followed hygienic menstrual practices faced reduced likelihood of sexual and emotional IPV. A study from Kenya [28] found that women who did not have access to safe and hygienic menstrual practices were more vulnerable to IPV. This is likely because communities where menstruation is a taboo, are most likely communities that are rigidly patriarchal and more prone to condoning IPV [29,30]. Conversely, women who are aware of hygienic menstrual practices and who use hygienic menstrual products are very likely to have increased exposure to media, access to education, and improved control over their bodies. Such women are likely to enter power-balanced intimate relationships which have reduced chances of IPV.

The study found that women who owned property were at a higher risk of IPV, particularly emotional IPV. A study from Pakistan [31], found that women owning property had more than twice the risk of IPV, and that husbands' controlling behaviours intensified when their wives started to own property. Controlling behaviour expresses as restricting mobility of the woman, being jealous of others around her, accusations of unfaithfulness [14]. These actions are potent to create stress and anxiety in the woman. It is likely that women in this study too were triggering the controlling attitudes in their husbands since they owned property. We also need to understand that the laws of the land, legislations that promote or prevent women's ownership of property impact the association between property ownership and IPV [32]. When legislations support inheritance by women, especially in case of her separation from their husbands, then women's property ownership is seen to lower risk of IPV. But in societies where property ownership is traditionally a male right, women's ownership of property is more likely to increase IPV [32]. The Agricultural Census (2015–16) India estimated that 14 percent of all agricultural land holdings belonged to women. Legislations in India support women's inheritance but, in several parts of the country, the cultural acceptance of this practice is likely to be still low. It is also possible that while women legally own land, decisions about the land, especially if it an agricultural plot, is taken by the males of the household [33]. This effectively implies that land ownership does not translate to women empowerment.

## Limitations

The cross-sectional nature of the survey data limits a causal association. The data does not let us assess the extent to which variables are interacting with each other. Further, this study did not look at determinants such as 'education of husband' or 'employment status of husband' owing to the lack of individual level data at the time of conducting this study. Lack of individual data may also lead to ecological fallacy, which would limit us from drawing conclusion at the individual level. Finally, due to the sensitive nature of IPV, there might have been an under-reporting among study participants.

## Future directions

Informed by these findings and the limitations of this study, future studies could focus on establishing causality using longitudinal observational studies using national datasets in order

to identify factors that positively associate with reduced IPV [34]. Obiagu points to evidence [20] that show significant reduction in IPV when intervention programs work with men and boys to change attitudes and perceptions; and, future studies could consider adopting mixed method studies such that big data is tempered with women's lived experiences of identifying IPV and other contextual factors. Investigating the connection between IPV and environmental factors, such as neighbourhood characteristics and resource accessibility, can offer useful insights [35]. Using GIS analysis to understand spatiotemporal dynamics of IPV and its association with programs on women empowerment can help us study how intervention contributes to a reduced burden. Following such a study, we can develop evidence-based interventions that are suited to the unique dynamics of various regions or communities by analysing spatial and temporal patterns.

## Conclusions

This study sought to understand how components of women empowerment associate with IPV, using data from the most recent round of the NFHS. The study found little reduction in IPV corresponding with improvement in women empowerment. However, the study adds to existing scholarship that demonstrate components of empowerment that associate with decreased burden of IPV. The three top factors associated with reduced IPV are women with household decision-making power, mobile phone ownership and hygienic menstrual practices. The findings of this study would inform future research and intervention that seek to strengthen these specific components of building women empowerment.

## Supporting information

**S1 Table. Questions asked to married women on their experience of different forms of IPV.**
(DOCX)

**S2 Table. Composite IPV scores (ranked from highest to lowest) versus the corresponding composite women empowerment score.**
(DOCX)

## Acknowledgments

The authors of this paper sincerely thank the support and solidarity of their peers Shagun Tuli, Rohini Dutta, Minal Shukla, Parth Tailor and Kranti Vora. They also sincerely acknowledge the amazing motivation and support of the Tuesday-NFHS Lab where updates on this paper were discussed on a weekly basis and feedback taken from the peers.

## Author Contributions

**Conceptualization:** Rakhi Ghoshal, Anita Gadgil, Nobhojit Roy.

**Data curation:** Priti Patil.

**Formal analysis:** Rakhi Ghoshal, Priti Patil, Prashant Bhandarkar.

**Methodology:** Priti Patil, Prashant Bhandarkar, Dnyaneshwar B. Kale.

**Resources:** Priyansh Nathani.

**Software:** Priti Patil, Prashant Bhandarkar.

**Supervision:** Nobhojit Roy.

**Visualization:** Dnyaneshwar B. Kale.

**Writing – original draft:** Rakhi Ghoshal.

**Writing – review & editing:** Rakhi Ghoshal, Priti Patil, Anita Gadgil, Priyansh Nathani, Nobhojit Roy.

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
