## [Decision Letter · Decision Letter 0]

7 Jun 2023

PONE-D-23-02776Women’s Household Decision-making, Mobiles Phones and Menstrual Hygiene associate with reduced IPV: findings from NFHS-5, IndiaPLOS ONE

Dear Dr. Roy,

Thank you for submitting your manuscript to PLOS ONE. After careful consideration, we feel that it has merit but does not fully meet PLOS ONE’s publication criteria as it currently stands. Therefore, we invite you to submit a revised version of the manuscript that addresses the points raised during the review process.

We look forward to receiving your revised manuscript.

Kind regards,

Hariom Kumar Solanki, M.D.

Academic Editor

PLOS ONE

4. We note that Figure 1 in your submission contain map images which may be copyrighted. All PLOS content is published under the Creative Commons Attribution License (CC BY 4.0), which means that the manuscript, images, and Supporting Information files will be freely available online, and any third party is permitted to access, download, copy, distribute, and use these materials in any way, even commercially, with proper attribution. For these reasons, we cannot publish previously copyrighted maps or satellite images created using proprietary data, such as Google software (Google Maps, Street View, and Earth). For more information, see our copyright guidelines: http://journals.plos.org/plosone/s/licenses-and-copyright.

Additional Editor Comments :

Dear authors

The study seems relevant and methodologically appropriate. However, please address the reviewer's comments for further consideration towards acceptance.

Thank you

Reviewers' comments:

Reviewer's Responses to Questions

**Comments to the Author**

1. Is the manuscript technically sound, and do the data support the conclusions?

Reviewer #1: Yes

Reviewer #2: Yes

2. Has the statistical analysis been performed appropriately and rigorously? 

Reviewer #1: No

Reviewer #2: Yes

3. Have the authors made all data underlying the findings in their manuscript fully available?

Reviewer #1: No

Reviewer #2: Yes

4. Is the manuscript presented in an intelligible fashion and written in standard English?

Reviewer #1: Yes

Reviewer #2: Yes

5. Review Comments to the Author

Reviewer #1: This is a very important subject area for social cohesion, however, the following concerns need attention by the authors.

The first statement of the abstract contradicts the last statements of the first and last paragraphs of the background.

The aim of the study is found in the fourth paragraph of the background but the authors have aim among others as a sub-heading in the method.

Lakh is used several times. It would help readers if the English meaning of Lakh could be put in brackets immediately after using it.

There are two sub-headings under the method that have “design”. The authors need to correct it.

The authors listed Microsoft Office Excel-2021, R Studio, and SPSS version 22.0 as the softwares used to analyse the data. There is a need for an explanation to be given as to how each software was used.

A summary of descriptive statistics of the variable could be given in the result section before the correlation matrix. This would throw more light on the manuscript.

If the authors attend to these concerns, the article will provide very relevant information for academics and the policymakers.

Reviewer #2: Reviewer Comments

Title

I will suggest that the authors reframe the topic as: Are Women’s’ Household decision-making, Mobile Phone Ownership, and Menstrual Hygiene associated with reduced IPV? Findings from NFHS-5, India.

Abstract

The background of the abstract simply ignored other important aspects of the work that were highlighted in the topic. Hence, the other parts such as mobile phone ownership and menstrual hygiene will need to be discussed unless they are simply measures of women’s household decision-making.

The method section of the abstract should include the sample size for this study, it’ll be important to show that.

Intimate Partner Violence should be included as a keyword.

Introduction

The last set of sentences on the relationship between employment and IPV risk (Pages 3-4) can be communicated in a single sentence.

Similar to the background section of the abstract, the introduction does not depict what the topic is communicating. The authors will need to decide whether this study is entirely focused on women’s empowerment and IPV, with mobile phone ownership and menstrual hygiene as measures of women’s empowerment or rewrite the whole introduction section to incorporate ideas on mobile phone ownership//menstrual hygiene and their links with IPV.

Method

Excellent description of the method section especially the statistical analyses employed in this study. However, the authors should endeavor to provide some more information on the study design, especially the type of sampling and how the NSFH study respondents were selected.

Results, Discussion, and Conclusion

The results were greatly presented, and the findings were discussed in light of similar findings in the literature.

It would be highly appropriate for the authors to offer suggestions for future studies in the context of the study.

6. PLOS authors have the option to publish the peer review history of their article (what does this mean?). If published, this will include your full peer review and any attached files.

Reviewer #1: No

Reviewer #2: No

---

## [Author Response · Author response to Decision Letter 0]

19 Jul 2023

EDITOR COMMENTS

All referencing and formatting style aligned with journal requirement. 

The required ethics statement has been added to the Method’s ethical considerations and data quality section, including a reference: [“Additional details on the protocol are available at https://dhsprogram.com/Methodology/Survey-Types/DHS.cfm.”] 

3. In your Data Availability statement, you have not specified where the minimal data set underlying the results described in your manuscript can be found. PLOS defines a study's minimal data set as the underlying data used to reach the conclusions drawn in the manuscript and any additional data required to replicate the reported study findings in their entirety. All PLOS journals require that the minimal data set be made fully available.

Upon re-submitting your revised manuscript, please upload your study’s minimal underlying data set as either Supporting Information files or to a stable, public repository and include the relevant URLs, DOIs, or accession numbers within your revised cover letter. 

Sorry for not being clear on this. All the Summary Fact Sheets and data used in the study is available on the DHS and NFHS website through their portal for registered users. We have made the edits on the DATA AVAILABILITY statement. (https://dhsprogram.com/data/http://rchiips.org/nfhs/factsheet_NFHS-5.shtml) 

4. We note that Figure 1 in your submission contain map images which may be copyrighted. All PLOS content is published under the Creative Commons Attribution License (CC BY 4.0), which means that the manuscript, images, and Supporting Information files will be freely available online, and any third party is permitted to access, download, copy, distribute, and use these materials in any way, even commercially, with proper attribution. For these reasons, we cannot publish 5previously copyrighted maps or satellite images created using proprietary data, such as Google software (Google Maps, Street View, and Earth). For more information, see our copyright guidelines: http://journals.plos.org/plosone/s/licenses-and-copyright.

We require you to either (1) present written permission from the copyright holder to publish these figures specifically under the CC BY 4.0 license, or (2) remove the figures from your submission. 

Figure-1 images have now been modified once more. 

We have done visualization using opensource technology R, which is a programming language and environment for Statistical computing and graphics. We used ggplot2 package of R for this purpose in the R studio for Windows. We believe this is in compliance with PLOS policy.

Figure-1 images have now been modified once more. We have done visualization using opensource technology R, which is a programming language and environment for Statistical computing and graphics. We used ggplot2 package of R for this purpose in the R studio for Windows. We believe this is in compliance with PLOS policy.

We have added 3 references (refs # 37, 38, 39). We have updated the Reference list as per the journal’s referencing style policy (Vancouver). 

REVIEWER#1 COMMENTS

1. The first statement of the abstract contradicts the last statements of the first and last paragraphs of the background. 

Yes indeed. Thank you. We have now edited the starting part of the Abstract to be synchronous with the Background. Background is also edited to be consistent. 

2. The aim of the study is found in the fourth paragraph of the background but the authors have aim among others as a sub-heading in the method. 

Yes, indeed we had duplicated it. We have now removed ‘aim’ from under Methods section. 

3. Lakh is used several times. It would help readers if the English meaning of Lakh could be put in brackets immediately after using it. 

We have decided to just use the numbers, since in millions, the numbers are less than even one million. 

4. There are two sub-headings under the method that have “design”. The authors need to correct it. 

Thank you for pointing that out. The redundant ‘Design’ word was removed from the Method section. 

5. The authors listed Microsoft Office Excel-2021, R Studio, and SPSS version 22.0 as the softwares used to analyse the data. There is a need for an explanation to be given as to how each software was used. 

The appropriate explanation was added to the Statistical Analysis section.

6. A summary of descriptive statistics of the variable could be given in the result section before the correlation matrix. This would throw more light on the manuscript. 

We have re-ordered the Results section to address the reviewer comment. The Result section now has the description (heatmap) first, followed by the table of Composite Score, followed by the Correlation Matrix. 

7. If the authors attend to these concerns, the article will provide very relevant information for academics and the policymakers. 

Thank you; we have tried our best to address all the concerns to the best of our capabilities. 

REVIEWER#2 COMMENTS

1. [Title:] I will suggest that the authors reframe the topic as: Are Women’s’ Household decision-making, Mobile Phone Ownership, and Menstrual Hygiene associated with reduced IPV? Findings from NFHS-5, India. 

We have now changed the title to make it broader. Detailed response related to the Title is also under point # 6 below. [Revised title is – Do empowered women have lowered risks of intimate partner violence in India? Evidence from National Family Health Survey- 5]

2. The background of the abstract simply ignored other important aspects of the work that were highlighted in the topic. Hence, the other parts such as mobile phone ownership and menstrual hygiene will need to be discussed unless they are simply measures of women’s household decision-making. 

We confirm that for this manuscript, mobile phone ownership and menstrual hygiene were our findings, not part of the aim. So, we have changed the title to be compatible with the Aim, and not Findings. Thus, we did not edit the Background (Abstract) to include mobile phone ownership or menstrual hygiene issues. We have left it at discussing women empowerment and its likely association with IPV, and that we are aiming to study the association for India. 

3. The method section of the abstract should include the sample size for this study, it’ll be important to show that. 

Agree with the reviewer. Details about sample size are now added.

4. Intimate Partner Violence should be included as a keyword. 

Sure, included now. 

5. [Introduction] The last set of sentences on the relationship between employment and IPV risk (Pages 3-4) can be communicated in a single sentence. 

Thank you. The sentences have been merged, and the references repositioned accordingly. 

6. Similar to the background section of the abstract, the introduction does not depict what the topic is communicating. The authors will need to decide whether this study is entirely focused on women’s empowerment and IPV, with mobile phone ownership and menstrual hygiene as measures of women’s empowerment or rewrite the whole introduction section to incorporate ideas on mobile phone ownership//menstrual hygiene and their links with IPV. 

This is indeed a significant discrepancy in the manuscript – thank you for highlighting it. 

Co-authors had a round of discussion and we realise that specifically mentioning mobile phones or menstrual hygiene in the title is not what we want to do. The paper set out to study the association with women empowerment and IPV (as stated in the aim) and found these variables as significant. Not the other way round. So, we have decided to change the title so that it is broader and reflects our primary question, and not the findings. So, we have not edited the Introduction to highlight literature on mobile phones and menstrual hygiene but kept the focus on empowerment and its broad indicators only, in terms of literature review in this Introduction. 

7. [Method] Excellent description of the method section especially the statistical analyses employed in this study. However, the authors should endeavor to provide some more information on the study design, especially the type of sampling and how the NSFH study respondents were selected. 

Thank you for your appreciation. We have now added the detailed sampling and the selection of sample size.

8. [Results, Discussion, and Conclusion] The results were greatly presented, and the findings were discussed in light of similar findings in the literature. 

Thank you. 

9. It would be highly appropriate for the authors to offer suggestions for future studies in the context of the study. 

Yes, we agree. We have added a paragraph on future research pathways at the end of the Discussion section.

---

## [Decision Letter · Decision Letter 1]

4 Sep 2023

PONE-D-23-02776R1Do empowered women have lowered risks of intimate partner violence in India? Evidence from National Family Health Survey-5PLOS ONE

Dear Dr. Roy,

Thank you for submitting your manuscript to PLOS ONE. After careful consideration, we feel that it has merit but does not fully meet PLOS ONE’s publication criteria as it currently stands. Therefore, we invite you to submit a revised version of the manuscript that addresses the points raised during the review process.

We look forward to receiving your revised manuscript.

Kind regards,

Hariom Kumar Solanki, M.D.

Academic Editor

PLOS ONE

Journal Requirements:

Additional Editor Comments:

Dear Authors

Thank you for submitting the revised manuscript, However, in view of reviewers comments and suggestions on the revised submission, the manuscript will require minor revision before further consideration.

Regards

Dr Hariom Kumar Solanki

Academic Editor

Reviewers' comments:

Reviewer's Responses to Questions

**Comments to the Author**

1. If the authors have adequately addressed your comments raised in a previous round of review and you feel that this manuscript is now acceptable for publication, you may indicate that here to bypass the “Comments to the Author” section, enter your conflict of interest statement in the “Confidential to Editor” section, and submit your "Accept" recommendation.

Reviewer #3: (No Response)

Reviewer #4: (No Response)

Reviewer #5: (No Response)

2. Is the manuscript technically sound, and do the data support the conclusions?

Reviewer #3: Partly

Reviewer #4: Partly

Reviewer #5: Yes

3. Has the statistical analysis been performed appropriately and rigorously? 

Reviewer #3: Yes

Reviewer #4: Yes

Reviewer #5: Yes

4. Have the authors made all data underlying the findings in their manuscript fully available?

Reviewer #3: Yes

Reviewer #4: Yes

Reviewer #5: Yes

5. Is the manuscript presented in an intelligible fashion and written in standard English?

Reviewer #3: Yes

Reviewer #4: Yes

Reviewer #5: Yes

6. Review Comments to the Author

Reviewer #3: 1.Kindly check last 3 keywords in MESH.If available then retain, otherwise delete

2.Last paragraph of the Background should be strong enough to identify gaps in the literature , rationale behind this study with relevance to IPV and women empowerment.

3.METHODS

i. kindly mention whether it is primary analysis or you analyzed an already conducted survey. Kindly mention about

sampling technique which will make it clear how that specific sample size was achieved.

ii. Justify why 03 Softwares were used for data analysis.

iii. Give a one liner which association will be called strongly linked

iv. What is level of confidence/level of significance adopted by you?

4.RESULTS

Kindly re-design table-2 giving info on composite scores, correlation coefficients and p-values.

Reviewer #4: 1. You have mentioned only female from each household were taken. Please specify how many households were taken in your study from the data of the NFHS survey. What was the selection criteria of the household from the total households. Specify sampling technique in detail from the total household to selection of sample from the sampling frame.

Reviewer #5: The work is a good addition to knowledge in the field of public health. Overall, the study will achieve study objective if observations made were taken- as applicable. Some sections of the results, discussion, limitation and references will need a revision. If not for data limitation, regression models would have improved the quality of study output.

7. PLOS authors have the option to publish the peer review history of their article (what does this mean?). If published, this will include your full peer review and any attached files.

Reviewer #3: No

Reviewer #4: No

Reviewer #5: **Yes: **ADEOYE, Philip Adewale

---

## [Author Response · Author response to Decision Letter 1]

10 Sep 2023

Section Reviewer comment Action taken Response by authors

Reviewer # 3

1

Keywords Kindly check last 3 keywords in MESH. If available then retain, otherwise delete Deleted So, we deleted them. 

Thank you for alerting us to this. 

The last three keywords (Spousal violence, Male backlash, Ownership of asset) are indeed not available as MeSH terms. 

2

Background Last paragraph of the Background should be strong enough to identify gaps in the literature, rationale behind this study with relevance to IPV and women empowerment. Added references and specifically identified gaps in literature. Thank you for the suggestion. 

We have edited the last paragraph of the Background, and bolstered the evidence on women empowerment, especially from India. 

We have identified specific gaps in this domain, as noted by other scholars, and pointed that out in Aims. The paper explicitly addresses those gaps. 

3

Methods kindly mention whether it is primary analysis or you analyzed an already conducted survey. Mentioned in ‘Design and Setting’ The analysis presented in our study is based on data from the 5th round of the National Family Health Survey (NFHS). We utilized existing NFHS-5 summary reports to conduct our analysis. 

Thus, the analysis is not a primary analysis; it is a secondary analysis of an already conducted survey. We have clearly mentioned this now. 

4

Methods Kindly mention about sampling technique which will make it clear how that specific sample size was achieved. Added in ms. Sure, thank you. 

The National Family Health Survey-5 (NFHS-5) uses a multistage sampling approach. It stratifies the country into states/union territories and district levels. Then, each stratum is subdivided into Primary Sampling Units based on rural-urban population. 

Appropriate details have been added to the section titled – Sample Size. 

5

Methods Justify why 03 Softwares were used for data analysis. Added explanation in ‘Statistical analysis’ section Thank you for the question. We opted to utilize three different software applications for our data analysis, each serving specific purposes. 

Microsoft Excel was selected for calculating the Dimension Index (DI) accurately. 

SPSS was employed for various statistical techniques.

R Studio was chosen for its proficiency in data visualization, ensuring that our results are presented in a clear and informative manner.

6

Methods Give a one liner which association will be called strongly linked Added reference to association in ‘Statistical analysis’ section Thank you; here it is. 

As indicated by Dancy and Reidy, Spearman's rank coefficient of correlation (ρ) with a value >= 0.40 was defined as a strong association. 

Necessary edits are now made in the Methods section to add this specificity to the manuscript. 

7

Methods What is level of confidence/level of significance adopted by you? Mentioned As mentioned in the manuscript, we have considered the standard 5% level of significance.

8

Results Kindly re-design table-2 giving info on composite scores, correlation coefficients and p-values. Title added to Table-2.

Declined Thank you for the suggestion. However, we would like to bring to your kind notice that Table-2 represents the correlation coefficient and p-values; since the headings or footnotes were missing, we have added them now in the revised version.

We would like to bring to your notice that in Table-2 the row contains women empowerment variables while column contains violence-related variables. Each of the row and column variables has its composite scores. Since table-2 is showing the association between row and column variables there is no way to show the granular data for individual variables when the variables are cross-tabulated. So, we are kindly declining this suggestion.

Reviewer # 4

9

Methods You have mentioned only female from each household were taken. Please specify how many households were taken in your study from the data of the NFHS survey. Added To comply with ethical requirements, only one eligible woman per household was selected to answer the questions in the domestic violence section. A total of 72,056 women from the same number of households responded to the domestic violence module. 

This information has now been added in the revised manuscript. 

10

Methods

 What was the selection criteria of the household from the total households. Added Thank you. As mentioned above, the sampling technique of NFHS has been added to the manuscript.

11

Methods Specify sampling technique in detail from the total household to selection of sample from the sampling frame. Added Thank you. As mentioned above, the sampling technique of NFHS has been added to the manuscript. 

Reviewer # 5

12

 The work is a good addition to knowledge in the field of public health. Overall, the study will achieve study objective if observations made were taken- as applicable. N/A Thank you for appreciating the sincerity of the work. We have greatly benefited from your supportive review. 

13

 Some sections of the results, discussion, limitation and references will need a revision. Addressed We have attended to each comment you made in track change on the ms. We have additionally ensured that sentences are clear, short and grammatically correct. We hope the revisions have now improved and strengthened the study. 

14 If not for data limitation, regression models would have improved the quality of study output. N/A We acknowledge that data limitations can indeed affect the performance and potential improvement of regression models. We have now mentioned the suggested limitations in the manuscript.

Comments by Reviewer # 5 directly on the manuscript 

Section Comment Action taken Response by authors 

15 

Results Figure title supposed to be below and not above. Done Thank you for pointing out that. The figure title is moved to the bottom of the figure now.

16 

Results Only decision-making shows a significantly strong corresponding reduction in the risk of IPV. Others (mobile phone ownership, menstrual hygiene, and land ownership) shows significantly moderate reduction in the risk of IPV. This might need a revision. Added explanation in ms. We would like to clarify that we have employed the association grouping criteria outlined by Dancey and Reidy. According to their guidelines, variables with Spearman's rank correlation coefficient (ρ) falling within the range of 0.30 to 0.39 are categorized as a moderate association, whereas variables with ρ values exceeding 0.40 are classified as strongly associated. We hope this explanation resolves any uncertainty.

17

Results “…. the component of owning property strongly linked to an increased risk of IPV….” 

--- Not strongly linked…but significantly moderate increased risk of IPV. This might also need a revision. Addressed We have added the details of the scale of association, and as per that, ρ value above 0.40 is considered as strong association.

18 “Employment and earning in cash linked to a slight increase in risk of IPV (0.10).” 

---- Not a significant result; and stating it might not be necessary since it also has a very weak correlation. Addressed Agreed. We will remove this line.

19 “When we disaggregated IPV along its three types (physical, sexual and emotional) …” 

------ You have at least 3 significant correlations for each of the types of IPV. You will need to report all significant correlates of IPV for a lay reader to understand the table. You will need to state the result as described in the preceding paragraph. Addressed We deeply value this reminder. We have made necessary edits to the part where we discuss Table-2 in the Results section. We have tried our best to put ourselves in the shoes of our readers and walk through all the significant findings systematically. Hope the reading of Table-2 is clearer now. 

20 In table – 2 …

----- You might also need to add “Correlation coefficient (p-value)” to each of the columns for easy comprehension of the result in the table by a lay reader. Addressed The table contains values representing the correlation coefficient and p-value. We have added the same in the footnote for more clarity.

21. Discussion Discussion should prioritize significant findings: which should include a statement of finding, comparison with other studies, reasons for the similarities or differences with other studies and/or reasons for the findings, and implications for public health.

discussion of women empowerment on sexual IPV should be done; since it is a significant finding. Content added as appropriate Thank you very much for this reminder. We have done this now, to the best of our abilities. We have repeated our top findings, and started each findings on a separate paragraph, and followed it up with discussion of literature showing similar or contrasting findings. We have tried to reflect on the socio-cultural reasons/contexts to the best of our knowledge, taking care to keep everything evidence based, and not slip into opinions. 

We have added a paragraph in the Discussion to reflect on women empowerment and likely negative association with sexual IPV. Thank you for drawing our attention to this gap. 

 In the part where we had written – hygienic menstrual practices associate with reduced IPV, especially sexual IPV 

------ Why the word “especially” since menstrual hygiene can also be said to be moderate correlates of reduced risk of both sexual and emotional IPV; according to the result. Rectified Yes, we acknowledge our error, and we have rectified it accordingly. We had missed ‘emotional IPV’ in this part and have now added it. 

 However, employment increases risk of IPV 

------ Please, this statement does not correlates with the result. Employment is not a significant correlate of overall, IPV for this study. Discussing this may not be necessary. Rectified Yes. We have removed this paragraph now. 

 Property ownership increased risk of IPV 

------- Any concordance or discordance with other studies? Added scholarship citations Yes, we have cited other literature to establish similarities. 

22

Limitations Lack of individual data may also lead to ecological fallacy. Which might be a limitation of the study in drawing conclusion at the individual level.

Due to the sensitive nature of IPV, there might have been an under-reporting of IPV among the study participants. Addressed / added Thank you for helping with strengthening the Limitation section. We have incorporated these. Further, we have added ‘Future Directions’ as a specific sub section after Limitations since there are quite a few suggestions that this study shares with readers. 

23.

References Three references were incomplete Completed Thank you. We have completed the entries now.

---

## [Decision Letter · Decision Letter 2]

13 Oct 2023

Does women empowerment associate with reduced risks of intimate partner violence in India?

Evidence from National Family Health Survey-5

PONE-D-23-02776R2

Dear Dr.  Roy,

We’re pleased to inform you that your manuscript has been judged scientifically suitable for publication and will be formally accepted for publication once it meets all outstanding technical requirements.

Kind regards,

Hariom Kumar Solanki, M.D.

Academic Editor

PLOS ONE

Additional Editor Comments (optional):

Reviewers' comments:

Reviewer's Responses to Questions

**Comments to the Author**

1. If the authors have adequately addressed your comments raised in a previous round of review and you feel that this manuscript is now acceptable for publication, you may indicate that here to bypass the “Comments to the Author” section, enter your conflict of interest statement in the “Confidential to Editor” section, and submit your "Accept" recommendation.

Reviewer #3: (No Response)

2. Is the manuscript technically sound, and do the data support the conclusions?

Reviewer #3: (No Response)

3. Has the statistical analysis been performed appropriately and rigorously? 

Reviewer #3: (No Response)

4. Have the authors made all data underlying the findings in their manuscript fully available?

Reviewer #3: (No Response)

5. Is the manuscript presented in an intelligible fashion and written in standard English?

Reviewer #3: (No Response)

6. Review Comments to the Author

Reviewer #3: (No Response)

7. PLOS authors have the option to publish the peer review history of their article (what does this mean?). If published, this will include your full peer review and any attached files.

Reviewer #3: No

---

## [Editor Report · Acceptance letter]

18 Oct 2023

PONE-D-23-02776R2 

Does women empowerment associate with reduced risks of intimate partner violence in India? Evidence from National Family Health Survey-5 

Dear Dr. Roy:

I'm pleased to inform you that your manuscript has been deemed suitable for publication in PLOS ONE. Congratulations! Your manuscript is now with our production department. 

Kind regards, 

on behalf of

Dr. Hariom Kumar Solanki 

Academic Editor

PLOS ONE